# Cellulose Fibers Hydrophobization via a Hybrid Chemical Modification

**DOI:** 10.3390/polym11071174

**Published:** 2019-07-11

**Authors:** Stefan Cichosz, Anna Masek

**Affiliations:** Lodz University of Technology, Institute of Polymer and Dye Technology, Faculty of Chemistry, Stefanowskiego 12/16, 90-924 Lodz, Poland

**Keywords:** cellulose fibers, moisture, maleic anhydride, solvent exchange, cellulose drying

## Abstract

The following article highlights the importance of an indispensable process in cellulose fibers (UFC100) modification which may change the biopolymer properties—drying. The reader is provided with a broad range of information considering the drying process consequences on the chemical treatment of the cellulose. This research underlines the importance of UFC100 moisture content reduction considering polymer composites application with the employment of a technique different than thermal treating. Therefore, a new hybrid chemical modification approach is introduced. It consists of two steps: solvent exchange (with ethanol either hexane) and chemical treatment (maleic anhydride—MA). With the use of Fourier-transform infrared spectroscopy (FT-IR), it has been proven that the employment of different solvents may contribute to the higher yield of the modification process as they cause rearrangements in hydrogen bonds structure, swell the biopolymer and, therefore, affect its molecular packing. Furthermore, according to the thermogravimetric analysis (TGA) and differential scanning calorimetry (DSC), the improvement in fibers thermal resistance was noticed, e.g., shift in the value of 5% temperature mass loss from 240 °C (regular modification) to 306 °C (while solvent employed). Moreover, the research was broadened with cellulose moisture content influence on the modification process—tested fibers were either dried (D) or not dried (ND) before the hybrid chemical treatment. According to the gathered data, D cellulose exhibits elevated thermal resistance and ND fibers are more prone to the MA modification. What should be emphasized, in the case of all carried out UFC100 treatments, is that a decrease in moisture contend was evidenced—from approximately 4% in case of thermal drying to 1.7% for hybrid chemical modification. This is incredibly promising considering the possibility of the treated fibers application in polymer matrix.

## 1. Introduction

Nowadays, cellulose fibers are the subject of various research studies [1,2,3,4,5,6,7,8,9,10,11,12]. In the following article, the stress is put on the moisture content influence on the mentioned biopolymer properties. Cellulose in an equilibrium state with the atmosphere always contains absorbed moisture which is in between 4 and 5 wt % [13]. This is not desirable considering polymer composites applications in which mentioned fibers are employed in the role of the filler. Elevated water content in cellulose powder may contribute to, e.g., poor polymer composite mechanical properties [14], weak filler-matrix adhesion [15], or depressed decomposition temperature of a product [16].

Therefore, cellulose powder requires drying before incorporation into the polymer matrix in order to obtain the best possible mechanical and thermal properties. According to the literature, water evolution upon the conventional thermal treatment might be divided into the three following stages [17]: (i) physical loss of water (<220 °C), (ii) chemical loss of water (220–550 °C), (iii) chemical loss of water in pyrolysis (>600 °C).

During the physical loss of water stage, physical desorption of moisture occurs and rather no chemical elimination of water is present before achieving approximately 220 °C [18]. What is more, until 200 °C is reached, H_2_O molecules are released from the cellulose surface [19]. It is also claimed that a temperature of 200 °C is the onset of water molecules evolution from cellulose according to chemical reactions [17].

By increasing the temperature, a chemical loss of water starts. At this moment, in dehydration reactions the anhydrocellulose is being formed [20,21]. These chemical processes may follow two different mechanisms: intermolecular (new covalent bonds are formed which causes increased reticulation and thermal stability of biopolymer) or intramolecular (elimination reactions from the hydroxyl groups to yield C=C double bonds; this phenomenon may contribute to the char composing benzene rings formation) [22]. What should be underlined is that dehydration reactions appear in the amorphous phase prior to the high depolymerization [23,24].

Around 600 °C and in oxygen-free conditions, the third stage is entered. Then the liquid fraction that consists mainly of water along with small amounts of acetylaldehyde, proprionaldehyde, furan, acetone, butanedione, and methanol is created [25]. At this stage various dehydration reactions appear [26].

Moreover, according to available information, it might be claimed that water content importantly promotes cellulose degradation. The hydrolysis reaction of the glucosidic bonds contributes to the incorporation of unstable acetalic chain-ends and the molecular weight decrease. Furthermore, taking into consideration the fact that water is a major product of the cellulose thermal decomposition, especially at the initial thermal treatment, the regarded process may be attributed to autocatalytic reactions [17,27,28,29,30].

From the material science point of view, the processes described above are mostly important considering the first and second water loss stages. The first of them concerns the problem of cellulose drying before powder incorporation into the polymer matrix. Then, the chemical loss of water (220–550 °C) step may help to understand the thermal behavior of prepared polymer composites regarding the initial moisture content in cellulose fibers.

This research is a follow-up of previously performed projects concerning the employment of cellulose fibers in polymer composites [31,32]. This time, the stress is mostly put on the water content in the analyzed biopolymer and its influence on the performed modification process. In this project, the main goal was to decline the water amount in cellulose fibers via various chemical treatments considering their potential employment in polymer composites. In this situation, the biopolymer moisture content is of a high importance while considering obtaining the material of the best possible properties [33,34,35,36]. Moreover, the new hybrid chemical treatment approach, which has been adapted from papermaking [37], is introduced. Therefore, regular chemical modification with maleic anhydride (MA) was broadened with the solvent exchange step that aims to decrease moisture content in cellulose fibers. The solvent exchange method and its impact on cellulose fiber properties has been widely described in literature [38,39,40]. What should be also emphasized, is that the MA treatment may contribute to the nanofibrils formation [41].

## 2. Materials and Methods

### 2.1. Materials

The Arbocel^®^ UFC100 Ultrafine Cellulose for Paper and Board Coating from J. Rettenmaier and Soehne (Rosenberg, Germany) was the type of cellulose used in this research. It is in a powder form (white, odorless) and its density is about 1.3 g/cm^3^. It is insoluble in water and fats. Nevertheless, this material exhibits a high water binding capacity (even at high temperatures and shearing forces). Its average fiber length is about 8 μm. the pH value varies between 5 and 7.5.

The provided cellulose fibers were modified with maleic anhydride (MA) provided by Sigma-Aldrich^®^ (Poznan, Poland). It is a white solid substance with a molecular mass of 98.06 g/mol and is commonly used in cellulose chemical modification. When dissolved in water it turns into maleic acid. Its melting point is somewhere in the region of 51–56 °C, the initial boiling point −200 °C and density is 1.48 g/cm^3^ at 20 °C. Vapor pressure of a compound is 0.2 hPa at 22 °C. The reagent was a commercial product of the highest purity available.

Solvents employed in the experiments, such as acetone (A), ethanol (99.9%) solution (E) and hexane (H), were bought from Chempur^®^ (Piekary Slaskie, Poland). All of them are colorless substances and their properties are listed in Table 1.

### 2.2. Cellulose Fibers Hybrid Chemical Modification

In this research a new hybrid chemical treatment approach is proposed. The regular surface modification of cellulose fibers was broadened with the solvent exchange in the filler, from water, to either ethanol or hexane. This approach was adapted from another study carried out by Vuoti et al. [37] which considered paper-making applications.

Solvent exchange was performed before and after the surface modification of cellulose fibers with MA in order to observe its effect on, subsequently, chemical modification of cellulose fibers and properties of polymer composite samples. Moreover, solvents of different polarity employment influence were examined (ethanol and hexane chosen on the basis of sorption experiments—data available in the Appendix A). According to sorption tests, ethanol was chosen as a polar solvent and as a non-polar one, hexane. They may exhibit different interactions with the natural filler. Furthermore, cellulose fibers were either dried (24 h, 100 °C; crystallizer 70 mm × 40 mm) or not dried before the hybrid chemical modification process in order to observe the effect of moisture content in the filler on the treatment yield. Additionally, some samples were modified only with the solvent so as to examine the ethanol/hexane impact on the cellulose properties. Figure 1 reveals all stages of hybrid modification process and Table 2 summarizes all of the performed modifications.

Solvent exchange: cellulose fibers were put into the flask and the solvent was poured (cellulose to solvent ratio—1:10). Then, such prepared dispersion was mixed with the dipol (400 rpm for ethanol and 1000 rpm for hexane) at room temperature. After 8 h, the mixing was switched off and the dispersion was left for the next 16 h in an ambient condition. When the time was up, the solvent was distilled in a vacuum rotary evaporator at 40 °C (60 rpm, 100 mbar in the case of ethanol and 250 mbar for hexane).

Modification with maleic anhydride: cellulose was put into the acetone and MA solution (cellulose to acetone ratio—1:10, cellulose to MA ratio—4:1) for 2 h (oil bath 40 °C, 60 r/min) in a rotary evaporator. When the process of stirring was finished, acetone was removed with the vacuum distillation process (oil bath 40 °C, initial pressure 200 mbar). Then, the sample was subjected to the heat in a vacuum oven at 100 °C, 440 mbar for 4 h.

Between the steps of solvent exchange and modification with MA, cellulose fibers were stored in a Binder^®^ oven (BINDER GmbH, Tuttlingen, Germany) at 70 °C (crystallizer 70 mm × 40 mm) and then after finishing the whole modification process at 40 °C (crystallizer 70 mm × 40 mm).

### 2.3. Fourier Transform Infrared Spectroscopy (FT-IR)

Cellulose fibers were dried for 24 h at 100 °C (Binder^®^ oven; crystallizer 70 mm × 40 mm) before being analyzed. Fourier transform infrared spectroscopy (FT-IR) absorbance spectra were investigated within the 4000–400 cm^−1^ range (64 scans, absorption mode). The experiment was performed with the use of a Thermo Scientific Nicolet 6700 FT-IR spectrometer (Thermo Fisher Scientific, Waltham, MA, USA) equipped with diamond Smart Orbit ATR sampling accessory.

### 2.4. Near Infrared Spectroscopy (NIR)

In this method, the near-infrared region of the electromagnetic spectrum was used. The measurements were carried out in the range of 10,000–4000 cm^−1^ in absorption mode—64 scans (Thermo Scientific, Nicolet 6700, Thermo Fisher Scientific, Waltham, MA, USA). Cellulose fibers were dried for 24 h at 100 °C (Binder^®^ oven; crystallizer 70 mm × 40 mm) before being analyzed.

### 2.5. Thermogravimetric Analysis (TGA)

Thermogravimetric analysis (TGA) was used in order to get acquainted with the thermal degradation process of cellulose fibers, detecting the mass loss as a function of raising temperature in the range of 25–600 °C (heating rate: 10 °C/min; Ar 60 cm^3^/min).

What is more, isothermal experiment was also carried out in order to observe the water evaporation process for neat cellulose fibers. The first step of the measurement was the heating of an oven from 25 to 100 °C (heating rate: 10 °C/min; Ar 60 cm^3^/min) and the second step was isothermal (100 °C) in an Ar atmosphere (60 cm^3^/min) for 120 min. The mass decrease was detected.

A Mettler Toledo TGA/DSC 1 STARe System (Mettler Toledo, Greifensee, Switzerland) equipped with Gas Controller GC10 (Mettler Toledo, Greifensee, Switzerland) was employed in this investigation. Activation energy values were calculated with the use of Broido’s method [42]. Cellulose fibers were dried for 24 h at 100 °C (Binder^®^ oven; crystallizer 70 mm × 40 mm) before being investigated.

### 2.6. Differential Scanning Calorimetry (DSC)

Cellulose fibers were dried for 24 h at 100 °C (Binder^®^ oven; crystallizer 70 mm × 40 mm) before being analyzed. Differential scanning calorimetry (DSC) investigation was performed in a temperature range from −20 to 200 °C (heating rate: 10 °C/min; Ar 60 cm^3^/min) prior to analysis of the water evaporation process establishing its enthalpy (ΔH) and temperature of the peak (T_peak_). Additionally, the Mettler Toledo TGA/DSC 1 STARe System equipped with a Gas Controller GC10 was employed.

### 2.7. Fischer Titration

Fischer titration was carried out with the use of Hydranal Solvent E and Hydranal Titrant 5E supplied by Fluka^®^ (Mettler Toledo, Greifensee, Switzerland). For each experiment, approximately 1.5 g of cellulose sample and 50 mL of Hydranal Solvent E were taken. The water content in the natural filler was established. The measurement was taken using the TitroLine Alpha (Schott^®^, Mainz, Germany) device. Cellulose fibers were dried for 24 h at 100 °C (Binder^®^ oven; crystallizer 70 mm × 40 mm) and, then, left in ambient conditions for 45 min before being investigated (it was impossible to carry out an accurate measurement immediately after taking the sample out of a dryer—too high water absorption rate).

## 3. Results and Discussion

### 3.1. FT-IR Investigation

Analyzing the FT-IR spectra of cellulose (Figure 2), one should notice some characteristic absorption bands, e.g., hydroxyl moieties at 3334 cm^−1^ [43] and 1030 cm^−1^ [44], C–H stretching vibration at 2896 cm^−1^ [45], and –COO at 1200–900 cm^−1^ [46]. Moreover, while cellulose fibers maleinization is performed, the peak at 1718 cm^−1^ occurs. It is assigned to –C=O bonds present in the structure of maleic acid [47]. Therefore, it is a perfect signal for tracking the modification process efficiency. Absorption bands assigned to chemical groups is tabularized in Table 3.

As mentioned above, the peak at 1718 cm^−1^ confirms the introduction of acid, and the signal at 1636 cm^−1^ indicates that the sample also contains vinyl groups [47]. What may be also noticed is the fact that in the case of fibers dried before the MA treatment, the absorption band at 1718 cm^−1^ is less visible than for not dried natural filler. It indicates some information about the yield of the carried-out treatment—UFC100 not dried before the MA grafting are modified in a more efficient way. This phenomenon is also confirmed for all of the performed chemical hybrid modifications (Figure 3).

Figure 3 also reveals that in the case of hybrid chemical modification, when MA treatment is combined with the solvent employment, more of the modifier is grafted in the case of the fibers which are not dried before the modification. Nevertheless, the effect of drying may overlap with the type of solvent employment, e.g., in the case of UFC100/ND/MA/1/E and UFC100/D/MA/1/E specimens, there is almost no difference in the intensities at 1720 cm^−1^. The phenomenon mentioned above may be explained by the nature of solvent exchange which is expected to make the amorphous areas on the fibrils more accessible. It also might lead to partial solubilization of fibrils into the reaction medium and loss of the initial microfibrillar structure [55].

According to Table 4, the highest intensities for absorption band at 1720 cm^−1^ are detected in the case of UFC100/ND/MA/0 (I_1720_ = 0.035), UFC100/ND/MA/2/H (I_1720_ = 0.024), and UFC100/ND/MA/2/E (I_1720_ = 0.022). Moreover, when having a closer look at tabularized values, one may observe the huge difference between the dried and not dried fibers, considering the intensity of the absorption band at 1720 cm^−1^, e.g., UFC100/ND/MA/0 with I_1720_ = 0.035 and UFC100/D/MA/0 with I_1720_ = 0.007. It may be also claimed that the different effect of the solvent employment for dried and not dried fiber modification may be observed. In the case of not dried fibers, solvents cause a decrease in the amount of grafted maleic anhydride. On the other hand, considering the dried fibers, employment of ethanol and hexane contributes to the improvement in the yield of the modification process.

Furthermore, considering the differences caused only be the hexane and ethanol treatment, according to Figure 4, some shifts between the peaks are visible, e.g., –OH stretching vibrations at 3334 cm^−1^ [43], C–H stretching vibrations at 2896 cm^−1^ [45], C–C/C–OH/C–H moieties at 1426 cm^−1^ [53], C–H/C=O/C–O–C bonds at 1314 cm^−1^ [44], and C–OH out-of-plane 558 cm^−1^ [48]. This might have been an effect of interaction changes between the fibers indicated by the solvent’s employment. Noticed shifts might be caused by the change of the hydrogen bonding energy in the system of internal and intermolecular interactions [56]. Therefore, the matter reacts to IR irradiation in a slightly altered way.

According to Figure 5 the potential water content in cellulose fibers may be analyzed, as the absorption band at 3332 cm^−1^ corresponds with H_2_O molecules absorbed by the natural filler, as well as –OH groups present in the cellulose structure [43]. It may be observed that in case of almost every performed modification the intensity at 3332 cm^−1^ is decreased in comparison with the neat, dried cellulose. Therefore, it may be expected that modified fibers would have lower capability of water bonding as the fibers have been hydrophobized via a performed treatment [57].

Surprisingly, also the C–H stretching bands situated at 2935 and 2900 cm^−1^ were found to be significantly impacted by the water uptake. Although, there is no clear chemical explanation for the observed changes, several hypotheses could bring elements of understanding. It could be inherent to the material which is known to be subjected to chemical intrinsic variability or be assigned to non-specific experimental variation [58].

### 3.2. NIR Analysis

Considering the Near Infrared Spectroscopy (NIR) spectra of cellulose fibers, some signals characteristic of this substance can be recognized, e.g., 8165 cm^−1^ (C–H stretching of the second overtone) [59], 7290 cm^−1^ (C–H stretching and C–H deformation vibrations) [59], 6472 cm^−1^ (O–H stretching of the first overtone) [59], and 4754 cm^−1^ (O–H stretching vibrations) [60]. Peaks assigned to chemical groups are tabularized in Table 5.

According to Figure 6, in the case of fibers not dried before the MA treatment, an increase in the intensity of 6706 cm^−1^ may be observed. At the same time, for fibers dried prior to chemical grafting, a decrease at this absorption band is noted. The mentioned signal is assigned to O–H stretching vibrations, but it also indicates the development of intramolecular hydrogen bonding [61]. This means that not only the chemical structure of the fibers is affected by the carried-out modification process, but also the intermolecular interactions between them. Similar tendencies are observed in the case of 5195 cm^−1^ (>C=O, adsorbed water) [59] and 4274 cm^−1^ (C=C bonds) [62].

Nevertheless, considering the data gathered in Figure 7 (Figure 7), there is no significant difference between the spectra of the fibers modified via a hybrid chemical approach and whether they were dried before the MA treatment or not. It may be also claimed that solvent exchanged does not incorporate further changes in the NIR spectra. Only slight variations in intensities are observed, mostly at 6706 cm^−1^ which may describe some changes in the interactions between the fibers via hydrogen bonds and the recombination of these bonds. Therefore, it is suspected that solvents are not affecting the chemical structure of cellulose but they are altering interactions between the natural fibers. This may be a consequence of solvent molecules adsorption on the fibers surface.

On the other hand, shifts visible in case of NIR spectra are stronger than the ones noticed with the use of FT-IR, e.g., 8165 cm^−1^ (C–H) [59], 6706 cm^−1^ (–OH, water) [61], and 5195 cm^−1^ (–OH water) [59]. This supports the idea of interaction changes between the fibers caused by the solvent incorporation, as well as the chemical modification process. Results obtained with the employment of the NIR technique are a perfect confirmation for changes that were observed previously in the FT-IR spectra. 

What is more, the absorption band at approximately 4754 cm^−1^ is assigned to the >C=O bond [60] which is present in the MA structure (Figure 6 and Figure 7). This signal may be employed to track the yield of the performed modification. It may be observed that the intensity of the peak at 4754 cm^−1^ is higher in the case of all treatments performed for UFC100 not dried before the MA treatment. Therefore, it may be said once more that, while fibers are not subjected to high temperatures prior to the chemical modification process, the reaction yield is improved.

### 3.3. TGA Investigation

According to the thermogravimetric analysis, some differences in thermal degradation process between modified cellulose samples are visible. Figure 8 reveals the general tendencies in investigated specimens. The primary mass loss is always assigned to the volatile matter release (up to 150 °C) [64,65]. The second decomposition step occurs at various temperatures with the initial point originated from 200 to 400 °C and is assigned to the cellulose thermal degradation [66].

First of all, MA treated natural fibers exhibit a lower mass decrease after the second decomposition step in comparison with UFC100/D/1440 (reference sample). Furthermore, it may be suspected that not dried and then modified cellulose fibers start to decompose at lower temperatures (approximately 200 °C) which is earlier than in the case of UFC100/D/MA/0 and according to the data available in different studies [67,68].

What should be undoubtedly emphasized is the similar shape of UFC100/D/1440 and UFC100/D/MA/0 curves. This may be caused by lesser grafting of MA on the surface of dried cellulose fibers in comparison with not dried ones, which has been described in the previous sections, as well as a consequence of the partial hornification process [69]. It is referred as the loss of swelling ability by cellulose fibers that results from drying and rewetting cycles [70].

Table 6 and Figure 9 reveal more data considering different cellulose modification approaches. On the basis of gathered information, it may be claimed that MA treatment lowers the thermal stability of cellulose fibers, as T_05%_ is decreased for all samples modified with MA, e.g., UFC100/ND/MA/0—T_05%_ = 240 °C, UFC100/D/MA/0—T_05%_ = 284 °C, UFC100/ND/1/H—T_05%_ = 304 °C, UFC100/D/1/H—T_05%_ = 306 °C, in comparison with specimens treated only with solvents, e.g., UFC100/ND/1/E—T_05%_ = 304 °C, UFC100/D/1/E—T_05%_ = 300 °C. T_05%_ is considered as a temperature at which 5% mass loss, which is more than water content in the cellulose samples, is detected. According to TGA data, water evaporation step mass loss varies between 1.2% and3.6% for MA treated UFC100/D and between 1.4% and 2.2% in case of MA modified UFC100/ND.

What is more, solvent treatment also caused an increase in T_05%_ in the case of MA treated specimens, e.g., for not dried cellulose fiber—from 240 °C (UFC100/ND/MA/0) up to 280 °C (UFC100/ND/MA/1/H), for dried natural filler—from 284 °C (UFC100/D/MA/0) up to 305 °C (UFC100/D/MA/1/H). In both cases, the highest improvement is observed in case of sample modified with hexane before the MA treatment. What is also interesting, in the case of the dried cellulose fibers T_05%_ varies between 284 and 306 °C (for MA modified samples T_05%_ = 284 and 305 °C) and for not dried ones—T_05%_ is between 240 and 304 °C (in case of MA modified samples T_05%_ = 240 and 280 °C). This proves the increased thermal stability of dried fibers, probably, mostly thanks to the hornification process [38].

According to the data gathered in Table 6 it may be also observed that the thermal degradation of cellulose fibers mostly varies among different modification approaches, until the temperature of 300–320 °C is reached. It is a temperature at which the mass loss of approximately 20% is detected. Then, the process begins to follow a similar path in the case of each sample (small differences considering T_50%_ and T_80%_ values). Nevertheless, it may be observed that most of the solvent treated fibers exhibit elevated T_90%_, e.g., UFC100/ND/MA/2/E (T_90%_ = 562 °C), UFC100/D/MA/1/H (T_90%_ = 560 °C), in comparison with the reference samples: UFC100/ND/MA/0 (T_90%_ = 496 °C) and UFC100/D/MA/0 (T_90%_ = 478 °C). Moreover, the residual weight in each TGA graph is assigned to the pyrolysis reaction products, which were not oxidized, as the experiment was performed in an argon atmosphere. A complexity of cellulose thermal degradation was widely described in literature [71,72].

Furthermore, in order to assess the differences between the modified samples in a more precise way, the activation energies of decomposition steps were calculated (Table 7). Presented data may be a useful tool in describing the water content in analyzed samples and corresponds well with the literature [73]. In general, the lower the
$E_{A_{1}}$
value, the less moisture is absorbed by the cellulose fibers [74]. Therefore, according to the presented results, UFC100/D/MA/2/H, UFC100/D/MA/0, UFC100/ND/MA/2/E, and UFC100/ND/MA/1/E are considered as the samples with the lowest moisture content.


What is more, it may be observed that in case of fibers dried before the MA treatment, the values are elevated in comparison with not dried cellulose, e.g., UFC100/ND/MA/0—$E_{A_{2}}$ = 76±1
kJ/mol, UFC100/D/MA/0—$E_{A_{2}}$ = 111±2 kJ/mol which is a consequence of chemical structure changes upon thermal treatment [75]. This corresponds well with the earlier noticed phenomenon, namely the higher thermal resistance of dried cellulose fibers [69]. Moreover, activation energy of cellulose fiber degradation is increased in the case of specimens modified only with the solvents (ethanol/hexane) in comparison with MA treated fibers, which also explains previously observed raised T_05%_ values in case of these samples.

### 3.4. DSC Analysis

Further results of a thermal analysis indicate some more information concerning the water content in analyzed cellulose fibers. Water evolution from cellulose occurs both physically through desorption and chemically by elimination reactions. Three distinct temperature regimes may be distinguished: (i) loss of absorbed water at low temperatures (<220 °C), (ii) loss of chemical water at moderate-to-high temperatures (220–550 °C), and (iii) loss of chemical water at high temperatures (>550 °C) [17]. Nevertheless, in this research the stress is put on the first stage of moisture evolution.

Figure 10 reveals the differences in DSC curves of MA treated cellulose fibers in comparison with the neat UFC100 dried for 24 h. It may be easily observed that enthalpy change (ΔH) calculated for the water evaporation phenomenon would be different for all of the presented samples which evidence the varied water content after each treatment. ΔH seems to be higher in case of fibers dried before the MA modification process. Observed variations might result from the changes in the length to diameter ratio during the carried out treatment—as referred in the literature [76].

On the basis of data gathered in Table 8 and Figure 11, it is exactly visible how the enthalpy change value connected to water evaporation process is changing among different cellulose fiber modification approaches. First of all, ΔH is elevated in the case of fibers dried before the chemical modification. Then, the enthalpy values vary from 35.5 to 76.9 J/g. In the case of not dried cellulose, it is between 40.0 and 56.7 J/g. 

What could be also observed is the shift of T_peak_ towards the higher values in case of only solvent treatment—T_peak_ of UFC100/ND/1/H, UFC100/ND/1/E, UFC100/D/1/H, and UFC100/D/1/E are in the region of 94 °C. This value is achieved by merely few samples, e.g., UFC100/D/MA/0, UFC100/D/MA/1/H, but most of the obtained results are at temperatures below 90 °C. Moreover, specimens treated with only hexane or ethanol exhibit also an improvement in ΔH in comparison with the other analyzed samples.

A higher enthalpy change value means more energy required to desorb the water from the cellulose fibers as dehydration heat increases considerably when the samples exhibit a raised moisture level [77]. This leads to the conclusion that, in general, fibers dried before the MA treatment exhibit higher moisture content levels in comparison with not dried cellulose specimens. Additionally, it may be also explained as a consequence of strong interactions between polymer chains which appear when the plasticizer (water) concentration decreases [78].

### 3.5. Fischer Titration Investigation

Fischer titration experiment [79,80] helped to establish the moisture content in cellulose fibers as an exact value given in a mass percentage. Figure 12 reveals differences in amount of water absorbed by all investigated samples.

In the literature, it is described that various chemical treatments [81,82,83] and solvent exchange [37] could alter cellulose fibers water uptake ability. According to Figure 12, it may be observed that each of the performed UFC100 treatments give better results in lowering the moisture level than the regular thermal approach (heating in a dryer)—UFC100/D. In general, MA treatment, whether it is performed on dried or not dried UFC100, contributes to the decrease in water amount absorbed by cellulose fibers.

Moreover, in the case of not dried UFC100, employment of different solvents causes a decrease in moisture content even to approximately 1.5% which is the lowest of detected values. A slightly different effect may be observed in the case of UFC100 dried before the MA treatment. Here, while ethanol or hexane are used, a more varied impact on absorbed water amount may be noticed.

On the other hand, when one compares an impact of hexane and ethanol employment in the modification process, it might be observed that the lowest moisture content values are noted in the case of the following specimens: UFC100/ND/MA/1/E (1.65%), UFC100/ND/1/E (1.79%), and UFC100/D/1/E (1.77%). In each of mentioned modification processes, ethanol was used. It is believed that due to the strong interactions between cellulose and ethanol, in comparison with hexane-fiber interaction forces, the biopolymer is swelled well and the fiber structure becomes more available to the solvent. As a consequence, a higher amount of water is desorbed and the fiber reactivity is improved—the effect of cellulose structure opening and reorganization of hydrogen bonds [38,40,84,85,86]. Therefore, it may be claimed that the employment of this solvent contributes to the UFC100 of a low moisture content and high obtained reactivity.

## 4. Conclusions

In this study, via hybrid chemical modification, cellulose fibers of decreased moisture content were produced. On the basis of the gathered results, it may be claimed that ethanol employment highly contributes to the lowering of water absorption ability of cellulose, e.g., UFC100/ND/MA/1/E (1.65%), UFC100/ND/1/E (1.79%), and UFC100/D/1/E (1.77%). Moreover, it was shown that cellulose fibers not dried before the hybrid chemical modification are more prone to MA grafting. The intensity of the absorption band at 1720 cm^−1^ was attributed to the bonded maleic anhydride, e.g., UFC100/ND/MA/0 with I_1720_ = 0.035 and UFC100/D/MA/0 with I_1720_ = 0.007 is a perfect confirmation for the UFC100 reactivity changes influenced by the drying process. Nevertheless, the effect of drying may be overlapped with the type of solvent employment, e.g., in the case of UFC100/ND/MA/1/E and UFC100/D/MA/1/E specimens there is almost no difference in the intensities at 1720 cm^−1^. Considering thermal properties, it may be said that MA treatment lowers the thermal stability of cellulose fibers, as T_05%_ is decreased for all samples modified with MA, e.g., UFC100/ND/MA/0—T_05%_ = 240 °C, UFC100/D/MA/0—T_05%_ = 284 °C, UFC100/ND/1/H—T_05%_ = 304 °C, UFC100/D/1/H—T_05%_ = 306 °C, in comparison with specimens treated with only solvents, e.g., UFC100/ND/1/E—T_05%_ = 304 °C, UFC100/D/1/E—T_05%_ = 300 °C. What is more, solvent treatment also caused an increase in T_05%_ in the case of MA treated specimens, e.g., for not dried cellulose fibers—from 240 °C (UFC100/ND/MA/0) up to 280 °C (UFC100/ND/MA/1/H), for dried natural filler—from 284 °C (UFC100/D/MA/0) up to 305 °C (UFC100/D/MA/1/H). Obtained results are incredibly promising considering the possibility of the treated fibers application in polymer matrixes, as sufficient hydrophobization level and decrease in water content have been only partially achieved in previous research studies [31,32]. 

## Figures and Tables

**Figure 1 polymers-11-01174-f001:**
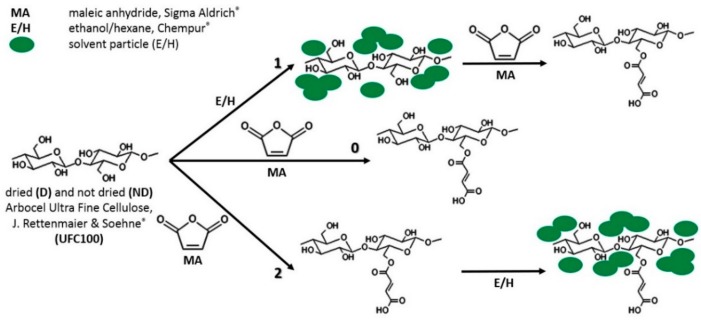
Hybrid chemical modification of cellulose fibers: Path 0—regular surface modification with maleic anhydride (MA); Path 1—solvent exchange before the surface modification with MA; Path 2—solvent exchange after the surface modification with MA.

**Figure 2 polymers-11-01174-f002:**
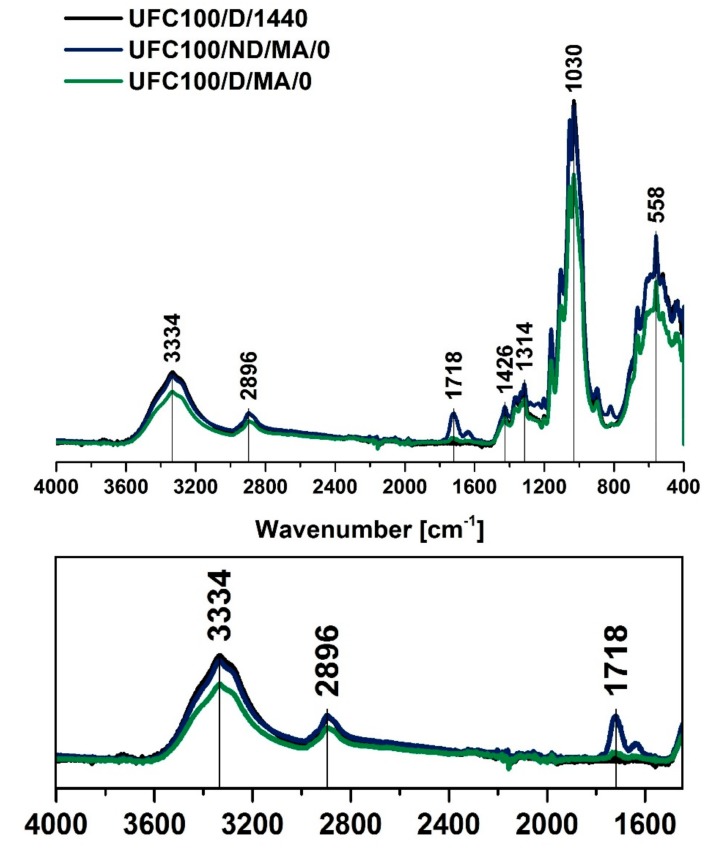
Fourier transform infrared spectroscopy (FT-IR) spectra of cellulose fibers modified via a regular approach (MA treatment) and unmodified. Characteristic absorption bands: 3334 cm^−1^ (O–H, water), 2894 cm^−1^ (C–H), 1718 cm^−1^ (C=O), 1200–900 cm^−1^ (O–H, C–O, –COO, CO–O–CO), and 558 cm^−1^ (C–OH, C–C).

**Figure 3 polymers-11-01174-f003:**
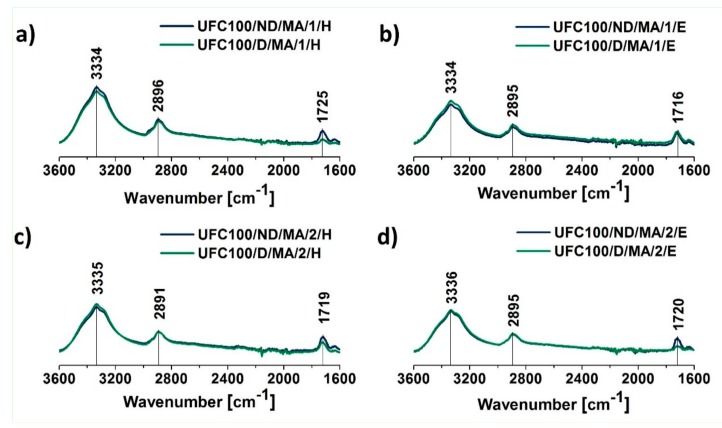
FT-IR spectra of cellulose fibers modified via a hybrid chemical modification: (**a**) solvent exchanged to hexane and then MA treated, (**b**) solvent exchanged to ethanol and then MA treated, (**c**) MA treated and then solvent exchanged to hexane, (**d**) MA treated and then solvent exchanged to ethanol. Characteristic absorption bands: 3334 cm^−1^ (O–H, water), 2894 cm^−1^ (C–H), and 1720 cm^−1^ (C=O).

**Figure 4 polymers-11-01174-f004:**
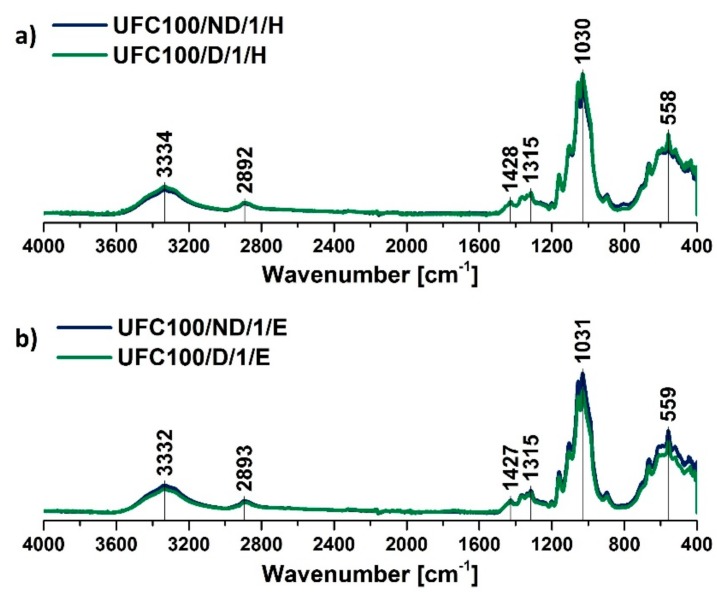
FT-IR spectra of solvent treated cellulose fibers: (**a**) solvent exchange to hexane, (**b**) solvent exchange to ethanol. Characteristic absorption bands: 3334 cm^−1^ (O–H, water), 2894 cm^−1^ (C–H), 1200–900 cm^−1^ (O–H, C–O, –COO, CO–O–CO), and 558 cm^−1^ (C–OH, C–C).

**Figure 5 polymers-11-01174-f005:**
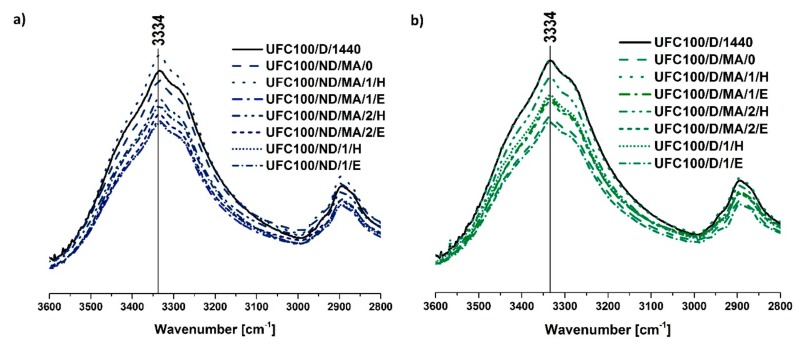
Comparison of the amount of water absorbed by modified cellulose fibers (3334 cm^−1^): (**a**) UFC100 not dried before the MA treatment, (**b**) UFC100 dried before the MA treatment.

**Figure 6 polymers-11-01174-f006:**
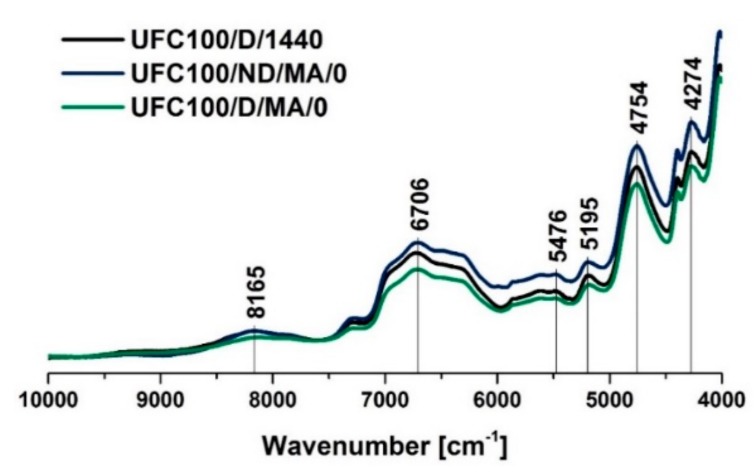
Near Infrared Spectroscopy (NIR) spectra of cellulose fibers modified via a regular approach (MA treatment) and unmodified. Characteristic absorption bands: 8165 cm^−1^ (C–H), 7290 cm^−1^ (C–H), 6772 cm^−1^ (O–H), 6706 cm^−1^ (O–H, water), and 4754 cm^−1^ (O–H).

**Figure 7 polymers-11-01174-f007:**
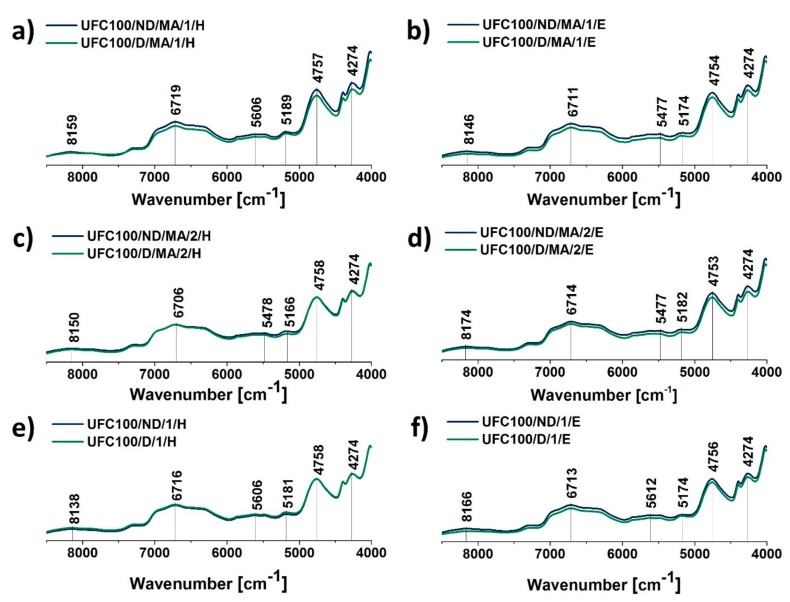
NIR spectra of cellulose samples modified via a hybrid chemical approach and solvent treatment: (**a**) solvent exchanged to hexane and then MA treated, (**b**) solvent exchanged to ethanol and then MA treated, (**c**) MA treated and then solvent exchanged to hexane, (**d**) MA treated and then solvent exchanged to ethanol, (**e**) solvent exchanged to hexane, (**f**) solvent exchanged to ethanol. Characteristic absorption bands: 8165 cm^−1^ (C–H), 7290 cm^−1^ (C–H), 6772 cm^−1^ (O–H), 6706 cm^−1^ (O–H, water), and 4754 cm^−1^ (O–H).

**Figure 8 polymers-11-01174-f008:**
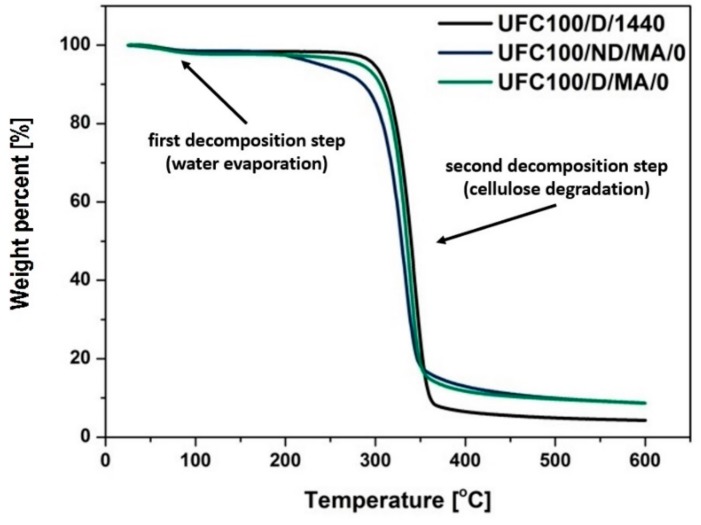
Comparison of MA treatment effect on thermal properties of dried and not dried cellulose fibers.

**Figure 9 polymers-11-01174-f009:**
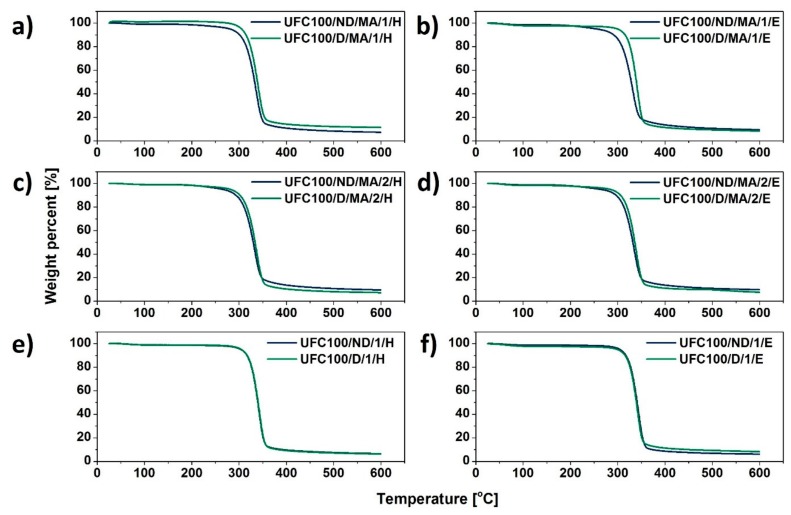
Thermogravimetric analysis (TGA) curves of cellulose samples modified via a hybrid chemical approach and solvent treatment: (**a**) solvent exchanged to hexane and then MA treated, (**b**) solvent exchanged to ethanol and then MA treated, (**c**) MA treated and then solvent exchanged to hexane, (**d**) MA treated and then solvent exchanged to ethanol, (**e**) solvent exchanged to hexane, (**f**) solvent exchanged to ethanol.

**Figure 10 polymers-11-01174-f010:**
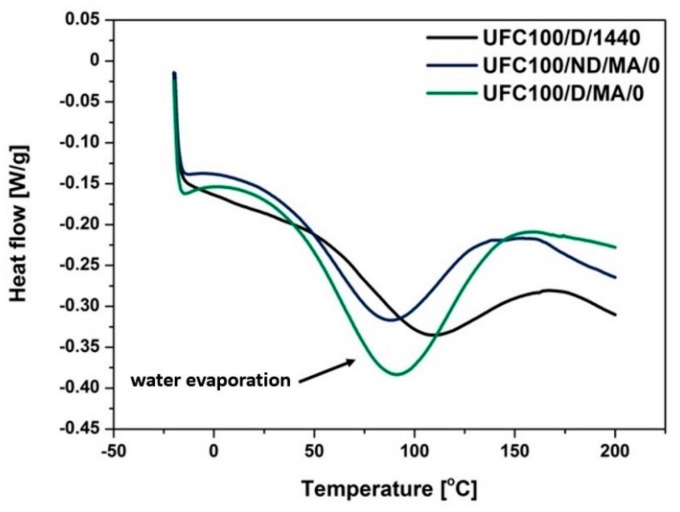
Comparison of cellulose fibers MA treatment effect on the water evaporation phenomenon according to the differential scanning calorimetry (DSC) curves.

**Figure 11 polymers-11-01174-f011:**
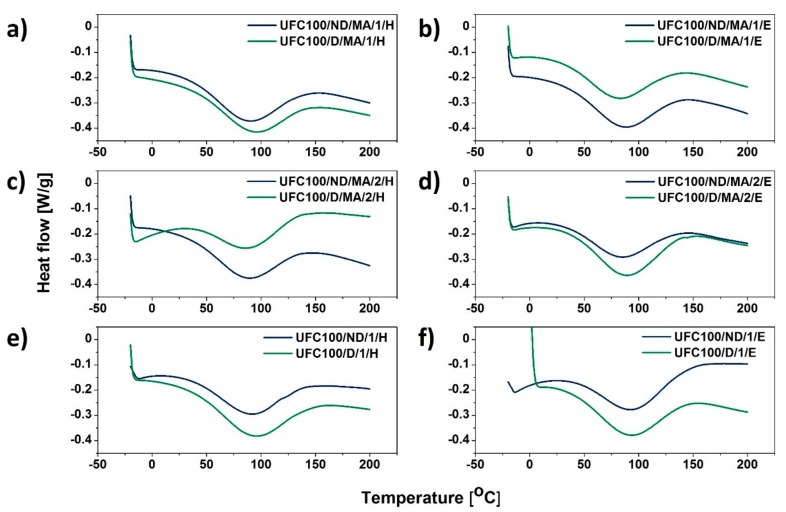
DSC curves of cellulose samples modified via a hybrid chemical approach and solvent treatment: (**a**) solvent exchanged to hexane and then MA treated, (**b**) solvent exchanged to ethanol and then MA treated, (**c**) MA treated and then solvent exchanged to hexane, (**d**) MA treated and then solvent exchanged to ethanol, (**e**) solvent exchanged to hexane, (**f**) solvent exchanged to ethanol.

**Figure 12 polymers-11-01174-f012:**
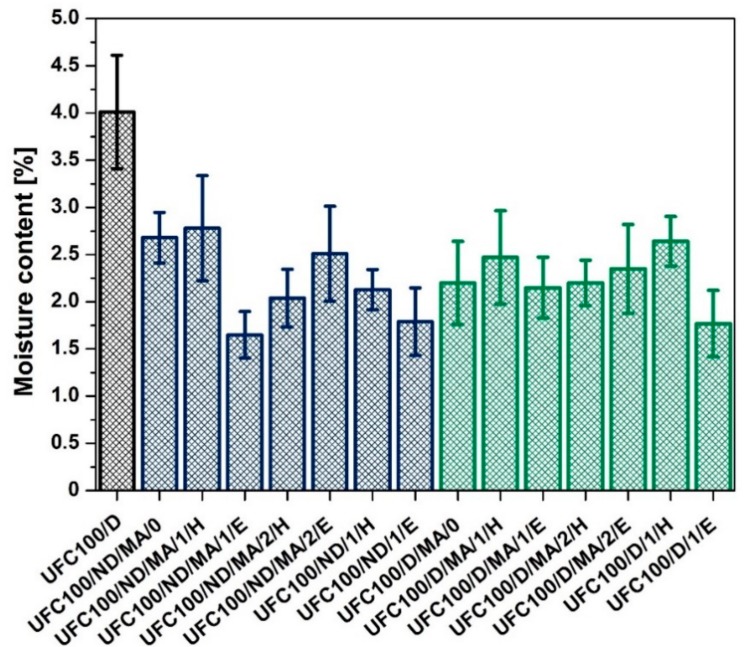
Moisture content in analyzed samples of cellulose fibers investigated with Fischer titration.

**Table 1 polymers-11-01174-t001:** Physical properties of solvents employed in experiments.

Property	Acetone	Ethanol (99.9%)	Hexane
Melting point (°C)	−95	−117	−95
Boiling point (°C)	55–57	78	68
Viscosity at 20 °C (mPas)	0.330	1.078	0.310
Density at 20 °C (g/cm^3^)	0.791	0.790	0.660
Vapor pressure at 20 °C (mbar)	233	60	160
Solubility in water (g/cm^3^)	yes	yes	0.1
Solubility in organic solvents	yes	yes	yes

**Table 2 polymers-11-01174-t002:** Summary of all performed cellulose fiber modifications.

Sample	UFC100 Dried before Modification (D)	UFC100 Not Dried before Modification (ND)	Solvent Exchange				MA Treated
			Path 1		Path 2		
			H	E	H	E	
UFC100/ND/MA/0	------	✔	------	------	------	------	✔
UFC100/ND/MA/1/H	------	✔	✔	------	------	------	✔
UFC100/ND/MA/1/E	------	✔	------	✔	------	------	✔
UFC100/ND/MA/2/H	------	✔	------	------	✔	------	✔
UFC100/ND/MA/2/E	------	✔	------	------	------	✔	✔
UFC100/ND/1/H	------	✔	✔	------	------	------	------
UFC100/ND/1/E	------	✔	------	✔	------	------	------
UFC100/D/MA/0	✔	------	------	------	------	------	✔
UFC100/D/MA/1/H	✔	------	✔	------	------	------	✔
UFC100/D/MA/1/E	✔	------	------	✔	------	------	✔
UFC100/D/MA/2/H	✔	------	------	------	✔	------	✔
UFC100/D/MA/2/E	✔	------	------	------	------	✔	✔
UFC100/D/1/H	✔	------	✔	------	------	------	------
UFC100/D/1/E	✔	------	------	✔	------	------	------

**Table 3 polymers-11-01174-t003:** Tabularized absorption bands assigned to the chemical groups.

Wavenumber (cm^−1^)	Chemical Group	Ref.
558	C–OH out-of-plane bending, C–C	[48]
1200–900	–OH, –COO	[46]
1100–1000	CO–O–CO	[49]
1030	C–O stretching vibration	[50]
1100	–OH	[44]
1158	C–O stretching vibration, C–O–C bridge	[51]
1245	–CH_3_	[52]
1314	C–O, C=O, C=C, COOH	[44]
1426	C–H bending of CH_2_	[53]
1635	–OH bending of adsorbed water, C=C	[54]
1718	C=O	[47]
2896	CH stretching vibration	[45]
3334	–OH, water	[43]

**Table 4 polymers-11-01174-t004:** Intensities at 1720 cm^−1^ for all performed MA modifications of cellulose.

Sample	I_1720_ (–)
UFC100/ND/MA/0	0.035
UFC100/ND/MA/1/H	0.020
UFC100/ND/MA/1/E	0.016
UFC100/ND/MA/2/H	0.024
UFC100/ND/MA/2/E	0.022
UFC100/D/MA/0	0.007
UFC100/D/MA/1/H	0.007
UFC100/D/MA/1/E	0.019
UFC100/D/MA/2/H	0.016
UFC100/D/MA/2/E	0.010

**Table 5 polymers-11-01174-t005:** Tabularized absorption bands assigned to the chemical groups.

Wavenumber (cm^−1^)	Chemical Group	Ref.
4754	–OH, C=O	[60]
5220–5150	–OH, water	[59]
6706	–OH, water, hydrogen bonds	[61]
4724	C=C	[62]
6772	–OH	[63]
7290	C–H	[59]
8165	C–H	[59]

**Table 6 polymers-11-01174-t006:** Temperatures of the mass loss; T_X%_—temperature at which the mass loss of x% is detected.

Sample	T_05%_ (°C)	T_10%_ (°C)	T_15%_ (°C)	T_20%_ (°C)	T_50%_ (°C)	T_80%_ (°C)	T_90%_ (°C)
UFC100/ND/MA/0	240	286	300	309	328	347	496
UFC100/ND/MA/1/H	280	301	310	315	333	343	418
UFC100/ND/MA/1/E	254	292	302	309	328	346	538
UFC100/ND/MA/2/H	265	294	304	310	330	348	546
UFC100/ND/MA/2/E	263	296	307	313	332	349	562
UFC100/ND/1/H	304	316	322	326	340	351	392
UFC100/ND/1/E	304	317	323	327	341	352	372
UFC100/D/MA/0	284	305	314	318	335	349	478
UFC100/D/MA/1/H	305	314	320	324	339	354	560
UFC100/D/MA/1/E	299	316	322	326	340	351	449
UFC100/D/MA/2/H	281	302	310	316	334	348	406
UFC100/D/MA/2/E	284	306	314	320	336	350	452
UFC100/D/1/H	306	317	323	327	340	352	383
UFC100/D/1/E	300	316	322	326	340	351	449

**Table 7 polymers-11-01174-t007:** Tabularized values of activation energy assigned to the decomposition steps calculated with the use of Broido’s method [42]; $E_{A_{1}}$—activation energy of water evaporation process,
$E_{A_{2}}$—activation energy of cellulose thermal degradation.

Sample	(kJ/mol)	(kJ/mol)
UFC100/ND/MA/0	73 ± 3	76 ± 1
UFC100/ND/MA/1/H	73 ± 2	103 ± 2
UFC100/ND/MA/1/E	69 ± 2	85 ± 1
UFC100/ND/MA/2/H	72 ± 2	85 ± 1
UFC100/ND/MA/2/E	66 ± 2	87 ± 1
UFC100/ND/1/H	77 ± 2	112 ± 1
UFC100/ND/1/E	73 ± 2	125 ± 3
UFC100/D/MA/0	54 ± 1	111 ± 2
UFC100/D/MA/1/H	75 ± 3	115 ± 2
UFC100/D/MA/1/E	74 ± 2	127 ± 3
UFC100/D/MA/2/H	45 ± 1	105 ± 2
UFC100/D/MA/2/E	73 ± 3	110 ± 2
UFC100/D/1/H	76 ± 2	125 ± 3
UFC100/D/1/E	74 ± 2	127 ± 3

**Table 8 polymers-11-01174-t008:** Tabularized water evaporation enthalpy values for all modified samples.

Sample	T_peak_ (°C)	ΔH (J/g)
UFC100/ND/MA/0	86	45.67
UFC100/ND/MA/1/H	88	56.68
UFC100/ND/MA/1/E	86	52.99
UFC100/ND/MA/2/H	87	52.04
UFC100/ND/MA/2/E	84	39.95
UFC100/ND/1/H	94	54.69
UFC100/ND/1/E	94	54.96
UFC100/D/MA/0	90	76.86
UFC100/D/MA/1/H	92	51.75
UFC100/D/MA/1/E	82	45.98
UFC100/D/MA/2/H	88	35.52
UFC100/D/MA/2/E	88	58.66
UFC100/D/1/H	94	64.44
UFC100/D/1/E	92	56.46

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
