# Peer review of "Cellulose Fibers Hydrophobization via a Hybrid Chemical Modification"

_polymers, 2019, doi:10.3390/polym11071174_

Round 1
Reviewer 1 Report
Manuscript entitled “Cellulose Fibres Hydrophobization via a Hybrid Chemical Modification” presented by authors is good work and can be accepted for publication after minor corrections. Few comments are as follows.
(1) Author need to improve quality of figure: 1.
(2) Author need to correct typos throughout the manuscript (like oC).
(3) Author need to incorporate few recent references related to Preparation and application of cellulosic or cellulose derivative materials; For example;
(a) materialstoday Volume, 21, Issue 7, September 2018, Pages 720-748 (b) Biomacromolecules 18 (8), 2333-2342 (c) ACS Sustainable Chemistry & Engineering 6 (3), 3279-3290 (d) ACS Applied Nano Materials 1 (8), 3969-3980 (e) Science China Technological Sciences 62 (6), 971-981 (f) Chemical Communications 49 (78), 8818-8820 (g) ACS Sustainable Chem. Eng.2019766140-6151 (h) Cellulose 25 (3), 1961-1973 (i) Industrial & Engineering Chemistry Research 56 (46), 13885-13893 (j) ACS Sustainable Chem. Eng.2019766140-6151 (k) J. APPL. POLYM. SCI. 2019, DOI: 10.1002/APP.47878 (l) Chem. Soc. Rev., 2018, 47, 2609—2679 (m) Chem. Rev. 2016, 116, 9305−9374 (n) Carbohydrate polymers 114, 339-343
(4) Why authors have chosen ethanol and either hexane for solvent exchange.
(5) How and why different solvents contribute to the higher yield of the modification process; author need to provide short expalination in the abstract part of the manuscript.
(6) Author need to include some interesting data in the abstract part to make it more interesting for the readers.
(7) What is the % of lignin present in the cellulose sample used by authors in their experiments because properties of cellulose is highly effected by lignin and hemicellulose contents.
(8) Author need to label FT-IR peaks with functional groups in the figure 3.
(9) TGA Fig 9, Mass change should be replaced by Wt%.
(10) What is the residual weight in each TGA graph author need to provide explanation.
(11) Why ethanol employment highly contributes to the lowering of water absorption ability of cellulose author need to provide proper explanation with suitable reference.
(12) Author need to compare their results with previously reported similar research.
Author Response
Institute of Polymer and Dye Technology
Technical University of Lodz
90-924 Lodz, ul Stefanowskiego 12/16, Poland
Tel.: +48 42 631 32 23, Fax: +48 42 636 25 43
July 8, 2019
Polymers MDPI
Dear Professor,
We are resubmitting our revised paper entitled Cellulose fibres modification via a hybrid chemical modification by, Stefan Cichosz, Anna Masek with a request to reconsider it for publication in Polymers.
We have carefully considered the Editor and Reviewers' comments. The manuscript was revised exactly according to these comments. The list of responses to the reviewer’s comments and corrections made in the manuscript is attached.
The manuscript has not been previously published, is not currently submitted for review to any other journal, and will not be submitted elsewhere before a decision is made by this journal.
For correspondence please use the following information:
corresponding author: Anna Masek
Institute of Polymer and Dye Technology
Technical University of Lodz
90-924 Lodz, ul Stefanowskiego 12/16, Poland
Tel.: +48 42 631 32 93
Fax: +48 42 636 25 43
e-mail: anna.masek@p.lodz.pl
Yours sincerely,
Ph. D., D.Sc. Anna Masek
Answers to reviewer #1 comments
Reviewer #1: Manuscript entitled “Cellulose Fibres Hydrophobization via a Hybrid Chemical Modification” presented by authors is good work and can be accepted
for publication after minor corrections. Few comments are as follows.
The comments are listed below.
Author need to improve quality of figure: 1.
Answer: Fig. 1 has been deleted according to the comment of Reviewer #3.
Author need to correct typos throughout the manuscript (like oC).
Answer: We have taken Reviewer’s comment into consideration. All typos
were corrected.
Author need to incorporate few recent references related to Preparation
and application of cellulosic or cellulose derivative materials; For example;
(a) materials today Volume, 21, Issue 7, September 2018, Pages 720-748
(b) Biomacromolecules 18 (8), 2333-2342 (c) ACS Sustainable Chemistry
& Engineering 6 (3), 3279-3290 (d) ACS Applied Nano Materials 1 (8), 3969-3980
(e) Science China Technological Sciences 62 (6), 971-981 (f) Chemical Communications 49 (78), 8818-8820 (g) ACS Sustainable Chem. Eng.2019766140-6151 (h) Cellulose 25 (3), 1961-1973 (i) Industrial & Engineering Chemistry Research 56 (46), 13885-13893 (j) ACS Sustainable Chem. Eng.2019766140-6151 (k) J. APPL. POLYM. SCI. 2019, DOI: 10.1002/APP.47878 (l) Chem. Soc. Rev., 2018, 47, 2609—2679 (m) Chem. Rev. 2016, 116, 9305−9374 (n) Carbohydrate polymers 114, 339-343
Answer: We are very grateful for gathering such a broad range of references that may enrich our article. All of mentioned research studies has been cited: Nowadays, cellulose fibres are the subject of various research studies [1-12]. In the following article the stress is put on the moisture content influence on the mentioned biopolymer properties.
Why authors have chosen ethanol and either hexane for solvent exchange.
Answer: We are grateful for this comment. Nevertheless, data is available
in Supplementary Materials. It is mentioned in 2.2. section - Moreover, solvents
of different polarity employment influence has been examined (ethanol and hexane chosen on the basis of sorption experiments – data available in a Supplementary Materials).
How and why different solvents contribute to the higher yield of the modification process; author need to provide short expalination in the abstract part of the manuscript.
Answer: This is a valuable advice. We have added this information in Abstract section: With the use of Fourier-transform infrared spectroscopy (FT-IR), it has been proved that the employment of different solvents may contribute to the higher yield
of the modification process as they causes rearrangements in hydrogen bonds structure, swell the biopolymer and, therefore, affect its molecular packing.
Author need to include some interesting data in the abstract part to make it more interesting for the readers.
Answer: We are grateful for this comment. The abstract part has been improved and some interesting results were presented, e.g., Furthermore, according
to the thermogravimetric analysis (TGA) and differential scanning calorimetry (DSC), the improvement in fibres thermal resistance has been noticed, e.g., shift in the value of 5% temperature mass loss from 240 °C (regular modification) to 306 °C (while solvent employed).
What is the % of lignin present in the cellulose sample used by authors in their experiments because properties of cellulose is highly effected by lignin and hemicellulose contents.
Answer: As it is mentioned in experimental section, the Arbocel® UFC100 Ultrafine Cellulose for Paper and Board Coating from J. Rettenmaier & Soehne was the type
of cellulose used in this research. It is in a powder form (white, odourless)
and its density is about 1.3 g/cm3. It is insoluble in water and fats. Nevertheless,
this material exhibits a high water binding capacity (even at high temperatures
and shearing forces). Its average fibre length is about 8 μm. pH value varies between
5-7.5.
It is a commercially bought cellulose. Nevertheless, we are not able to present the data considering the lignin and hemicellulose content.
Author need to label FT-IR peaks with functional groups in the figure 3.
Answer: We are thankful for Reviewer’s comment. However, data considering
the assignment of the peaks presented in Fig. 3 is available in Table 3. There
are the exact values given. The table has been improved.
TGA Fig 9, Mass change should be replaced by Wt%.
Answer: The mistake has been corrected.
What is the residual weight in each TGA graph author need to provide explanation.
Answer: We thank Reviewer for paying attention to this problem. Some more information has been given, e.g., Moreover, the residual weight in each TGA graph
are pyrolysis reaction products, which were not oxidized, as the experiment
has been performed in an argon atmosphere. A complexity of cellulose thermal degradation was widely described in literature [58,59].
Why ethanol employment highly contributes to the lowering of water absorption ability of cellulose author need to provide proper explanation with suitable reference.
Answer: We are thankful for Reviewer’s comment. The explanation has been given: In each of mentioned modification process ethanol was used. It is believed that due
to the strong interactions between cellulose and ethanol, in comparison with hexane-fibre interaction forces, the biopolymer is swelled well and the fibre structure becomes more available to the solvent. As a consequence, higher amount of water is desorbed and the fibre reactivity is improved – the effect of cellulose structure opening and reorganization of hydrogen bonds [26,28,71–73]. Therefore, it may be claimed that the employment of this solvent contributes to the UFC100 of a low moisture content obtaining.
Author need to compare their results with previously reported similar research.
Answer: As presented chemical modification is a new approach, unfortunately,
it is hard to compare the results with different research studies, especially considering the water content. Nevertheless, when possible, the comparison is given,
e.g., Furthermore, in order to assess the differences between the modified samples
in a more precise way, the activation energies of decomposition steps have been calculated (Table 7). Presented data may be a useful tool in describing the water content in analysed samples and goes in a well correspondence with the literature [60] or Literature says that various chemical treatments [68–70] and solvent exchange [25] could alter cellulose fibres water uptake ability.

Reviewer 2 Report
This is a carefully done study which merits publication. In order to emphasize the significance of this work to readers, the authors might want to add the meaning of the obtained data and make adequate revisions. The following is a list of my comments, including those points and other noticed ones.
What was the reaction efficiency (yield) of MA? Is the yield a realistic value as a practical process?
Is the degree of the decrease in water content (hydrophobization) obtained by MA modification practically meaningful? Has the required hydrophobization been achieved for the composites which the authors dealt with previously?
In addition to hydrophobization, MA modification has merits such as promotion of nanofibre formation. How about citing recent literature (e.g., ACS Macro Lett. 2015, 4, 80-83)?
Have you confirmed that free MA has been completely removed after modification with MA? At what stage of the procedure described in 2.2 will MA be removed?
In Fig. 3, since the spectra are displayed overlapping, it is difficult to read the difference among them.
L193: died -> dried
LL98-LL99 and LL230-231: Unnecessary line breaks are found.
Colloquialisms are found occasionally. How about asking a specialist for reviewing scientific English writing?
Author Response
Institute of Polymer and Dye Technology
Technical University of Lodz
90-924 Lodz, ul Stefanowskiego 12/16, Poland
Tel.: +48 42 631 32 23, Fax: +48 42 636 25 43
July 8, 2019
Polymers MDPI
Dear Professor,
We are resubmitting our revised paper entitled Cellulose fibres modification via a hybrid chemical modification by, Stefan Cichosz, Anna Masek with a request to reconsider it for publication in Polymers.
We have carefully considered the Editor and Reviewers' comments. The manuscript was revised exactly according to these comments. The list of responses to the reviewer’s comments and corrections made in the manuscript is attached.
The manuscript has not been previously published, is not currently submitted for review to any other journal, and will not be submitted elsewhere before a decision is made by this journal.
For correspondence please use the following information:
corresponding author: Anna Masek
Institute of Polymer and Dye Technology
Technical University of Lodz
90-924 Lodz, ul Stefanowskiego 12/16, Poland
Tel.: +48 42 631 32 93
Fax: +48 42 636 25 43
e-mail: anna.masek@p.lodz.pl
Yours sincerely,
Ph. D., D.Sc. Anna Masek
The comments are listed below.
.
Answers to reviewer #2 comments
Reviewer #2: This is a carefully done study which merits publication. In order
to emphasize the significance of this work to readers, the authors might want to add
the meaning of the obtained data and make adequate revisions. The following is a list
of my comments, including those points and other noticed ones.
The comments are listed below.
What was the reaction efficiency (yield) of MA? Is the yield a realistic value
as a practical process?
Answer: We are thankful for Reviewer’s comment. The exact value of the reaction yield is hard to be assessed. However, the amount of grafted MA is sufficient in order to observe the changes in the cellulose-filled polymer composite samples –
this is a topic of a further research study and follow-up article.
Is the degree of the decrease in water content (hydrophobization) obtained by MA modification practically meaningful?
Answer: It is of a huge importance considering the polymer-based composite applications. In general, high water content causes the decrease of both mechanical and thermal properties.
Has the required hydrophobization been achieved for the composites
which the authors dealt with previously?
Answer: No, in order to underline it, the conclusions section has been changed: Obtained results are incredibly promising considering the possibility of the treated fibres application in polymer matrix, as sufficient hydrophobization level and decrease in water content have been only partially achieved in previous research study [31,32].
In addition to hydrophobization, MA modification has merits such as promotion of nanofibre formation. How about citing recent literature (e.g., ACS Macro Lett. 2015, 4, 80-83)?
Answer: We are thankful for Reviewer’s comment and sharing the valuable information considering MA treatment. The article has been mentioned in the text: What should be also emphasises, the MA treatment may contribute to the nanofibrils formation [41].
Have you confirmed that free MA has been completely removed after modification with MA? At what stage of the procedure described in 2.2 will MA be removed?
Answer: In case of the modification performed according to the first path, the MA has been not removed as each drying-wetting cycle influence the properties of the fibre and it would be hard to compare the effect of two performed approaches. In case of the second path reaction, the MA residues have been washed during the solvent exchange to hexane, either ethanol.
In Fig. 3, since the spectra are displayed overlapping, it is difficult to read
the difference among them.
Answer: We thank for drawing our attention to this problem. Fig. 3 (present Fig. 2) has been improved.
L193: died -> dried
Answer: We are terribly sorry for this typo. The mistake has been corrected.
LL98-LL99 and LL230-231: Unnecessary line breaks are found.
Answer: We are grateful for Reviewer’s comment. Nevertheless, we cannot see
the line breaks in mentioned lines. It could be due to different versions of Microsoft Office.
Colloquialisms are found occasionally. How about asking a specialist
for reviewing scientific English writing?
Answer: We are thankful for Reviewer’s advice. The whole text has been checked once more in order to improve the writing style.

Reviewer 3 Report
1. make space
<220_°C
2. Figure 1 is not necessary
3. According to the Table 4
remove "the"
4. I cannot figure out the dot graphs in Figure 6
It is an online journal. There is no limitation on the number of colors.
You can use different colors for each graph.
Author Response
Institute of Polymer and Dye Technology
Technical University of Lodz
90-924 Lodz, ul Stefanowskiego 12/16, Poland
Tel.: +48 42 631 32 23, Fax: +48 42 636 25 43
July 8, 2019
Polymers MDPI
Dear Professor,
We are resubmitting our revised paper entitled Cellulose fibres modification via a hybrid chemical modification by, Stefan Cichosz, Anna Masek with a request to reconsider it for publication in Polymers.
We have carefully considered the Editor and Reviewers' comments. The manuscript was revised exactly according to these comments. The list of responses to the reviewer’s comments and corrections made in the manuscript is attached.
The manuscript has not been previously published, is not currently submitted for review to any other journal, and will not be submitted elsewhere before a decision is made by this journal.
For correspondence please use the following information:
corresponding author: Anna Masek
Institute of Polymer and Dye Technology
Technical University of Lodz
90-924 Lodz, ul Stefanowskiego 12/16, Poland
Tel.: +48 42 631 32 93
Fax: +48 42 636 25 43
e-mail: anna.masek@p.lodz.pl
Yours sincerely,
Ph. D., D.Sc. Anna Masek
The comments are listed below.
Answers to reviewer #3 comments
The comments are listed below.
Make space <220_°C
Answer: The mistake has been corrected.
Figure 1 is not necessary
Answer: We are thankful for Reviewer’s comment. Fig. 1 has been removed.
According to the Table 4 remove "the"
Answer: The text has been checked once more and all of the found mistakes corrected.
I cannot figure out the dot graphs in Figure 6 It is an online journal. There
is no limitation on the number of colors. You can use different colors for each graph.
Answer: We cannot agree more with the Reviewer’s comment. However, according
to the FT-IR spectra, water content cannot be assessed precisely – the method
is not accurate enough. It may be only suspected that the water content is higher/lower. Moreover, throughout the whole article everything connected with the dried fibres
is marked with green colour and in case of not dried ones – blue. Therefore, we have decided to prepare a figure like this, so as not to disturb the reader with suddenly appearing different colours. The aim of this figure is to show the possibly lower moisture content in case of modified fibres and not the exact values. Nevertheless,
we have changed the figure a little bit to make it more readable.
